# Deep Ensemble of Weighted Viterbi Decoders for Tail-Biting Convolutional Codes

**DOI:** 10.3390/e23010093

**Published:** 2021-01-10

**Authors:** Tomer Raviv, Asaf Schwartz, Yair Be’ery

**Affiliations:** School of Electrical Engineering, Tel-Aviv University, Tel-Aviv 6997801, Israel

**Keywords:** deep learning, error correcting codes, viterbi, machine learning, ensembles, tail-biting convolutional codes

## Abstract

Tail-biting convolutional codes extend the classical zero-termination convolutional codes: Both encoding schemes force the equality of start and end states, but under the tail-biting each state is a valid termination. This paper proposes a machine learning approach to improve the state-of-the-art decoding of tail-biting codes, focusing on the widely employed short length regime as in the LTE standard. This standard also includes a CRC code. First, we parameterize the circular Viterbi algorithm, a baseline decoder that exploits the circular nature of the underlying trellis. An ensemble combines multiple such weighted decoders, and each decoder specializes in decoding words from a specific region of the channel words’ distribution. A region corresponds to a subset of termination states; the ensemble covers the entire states space. A non-learnable gating satisfies two goals: it filters easily decoded words and mitigates the overhead of executing multiple weighted decoders. The CRC criterion is employed to choose only a subset of experts for decoding purpose. Our method achieves FER improvement of up to 0.75 dB over the CVA in the waterfall region for multiple code lengths, adding negligible computational complexity compared to the circular Viterbi algorithm in high signal-to-noise ratios (SNRs).

## 1. Introduction

Wireless data traffic has grown exponentially over recent years with no foreseen saturation [1]. To keep pace with connectivity requirements, one must carefully attend available resources with respect to three essential measures: reliability, latency, and complexity. As error correction codes (ECC) are well renowned as means to boost reliability, the research of practical schemes is crucial to meet demands.

One family of ECC that has had great impact on wireless standards is the convolutional codes (CC). Specifically, tail-biting convolutional codes (TBCC) [2] were incorporated in the 4G Long-Term Evolution (LTE) standard [3], and they are also considered for 5G hybrid turbo/LDPC code-based frameworks [4].

The major practical difference between CC and TBCC lies in the termination constraint. Conventional CC encoding appends zeros bits to impose zero states; TBCC encoding requires no additional bits, avoiding the rate loss. Due to this rate loss aversion, TBCC dominates classical CC in the short-length regime: They achieve the minimum distance bound for a specified length block codes [5]. Our work focuses on improving decoding performance of short length TBCC due to their significance.

Despite having great potential, the optimality of TBCC with respect to the reliability, latency, and complexity measures is not yet guaranteed. For example, TBCC suffer from increased complexity in the maximum-likelihood decoding as the initial state is unknown. Under TBCC encoding the Viterbi algorithm (VA) [6] is not the maximum-likelihood decoder (MLD); the MLD operates by running a VA per initial state, outputting the most likely decoded codeword. Clearly, the complexity grows as the number of states.

To bridge the high complexity gap, several suboptimal and reduced complexity decoders have been proposed. Major works include the circular Viterbi algorithm (CVA) [7] and the wrap-around Viterbi algorithm (WAVA) [8]. Both methods utilize the circular nature of the trellis: They apply VA iteratively on the sequence of repeated log-likelihood ratios (LLR) values computed from the received channel word. The more repetitions employed, the lower the error rate is, yet at a cost of additional latency. Short-length codes require the additional repetitions, as indicated in [7]: “it is very important that a sufficient number of symbol times be allowed for convergence. If the number is too small…the path chosen in this case will be circular but will not be the maximum likelihood path.”

Another common practice is to employ a list decoding scheme, for instance, the list Viterbi algorithm (LVA), along with cyclic redundancy check (CRC) code [9,10,11]. According to this scheme, a list of the most likely decoded codewords, rather than a single codeword, is computed in the forward pass of the VA. The minimal path metric codeword that satisfies the CRC criterion is output. Both the decrement in error rate and the increase in complexity are proportional to the list size.

The additional repetitions and list size result in complexity overhead for short TBCC decoding. To mitigate this overhead, one may take a novel approach, rooted in a data-driven field: the machine learning (ML) based decoding.

Still a growing field, ML-based decoding attempts to bridge the gap between simple analytical models and the nonlinear observable reality. Contemporary literature is split between two different model choices: model-free and model-based. Model-free works include those in [12,13,14], which leverage on state-of-the-art (SOTA) neural architectures with high neuronal capacity (i.e., ones that are able to implement many functions). On the other hand, under model-based approaches [15,16,17,18], a classical decoder is assigned learnable weights and trained to minimize a surrogate loss function. This approach suffers from high inductive bias due to the constrained architecture, leading to limited hypothesis space. Nonetheless, it generalizes better to longer codes than the model-free approach: Empirical simulations show that unrealistic fraction of codewords from the entire codebook must be fed to the network to achieve even moderate performance (see Figure 7 in [12]). One notable model-based method by Shlezinger et al. is the ViterbiNet [19]. This method compensates for nonlinearity in the channel with expectation-maximization clustering; an NN is utilized to approximate the marginal probability. This method holds great potential for dealing with nonlinear channels.

One recent innovation, referred to as the ensemble of decoders [20], combined the benefits of model-based approach with the list decoding scheme. This ensemble is composed of learnable decoders, each one called an expert. Each expert is responsible for decoding channel words that lie in a unique part of the input space. A low-complexity gating function is employed to uniquely map each channel word to its respective decoder. The main intuition behind this divide-and-conquer approach is that combination of multiple *diverse* members is expected to perform better than all individual basic algorithms that compose the ensemble. The paper shows this approach can achieve significant gains: Up to 1.25 dB on the benchmark.

The main contributions of this paper are the innovation of the model-based weighted circular Viterbi algorithm (WCVA) and its integration in the gated WCVAE, a designated ensemble of WCVA decoders, accompanied by a gating decoder. Next, we elaborate on the following major points.

**WCVA**—A parameterized CVA decoder, combining the optimality of the VA with a data-driven approach in Section 3.1. Viterbi selections in the WCVA are based on the sums of weighted path metrics and the relevant branch metrics. The magnitude of a weight reflect the contribution of the corresponding path or branch to successful decoding of a noisy word.**Partition of the channel words space**—We exploit the domain knowledge regarding the TBCC problem and partition the input space to different subsets of termination states in Section 3.3; Each expert specializes on codewords that belong to a single subset.**Gating function**—We reinforce the practical aspect of this scheme by introducing a low-complexity gating that acts as a filter, reducing the number of calls to each expert. The gating maps noisy words to a subgroup of experts based on the CRC checksum (see Section 3.4).

Simulations of the proposed method on LTE-TBCC appear in Section 4.

Please see Appendix A for introduction video and python code.

## 2. Background

### 2.1. Notation

Boldface upper-case and lower-case letters refer to matrices and vectors, respectively. Probability mass functions and probability density functions are denoted with P(·). Subscripts refer to elements, with the *i*th element of the vector x symbolized as $x_{i}$, while superscripts in brackets, e.g., $x^{\left(j\right)}$, index the *j*th vector in a sequence of vectors. A slice of a vector $\left(x_{i},\ldots ,x_{j}\right)$ is denoted by $x_{i;j}$. At last, ${\left(\cdot \right)}^{T}$ is for the transpose operation and ‖·‖ is for the L1 norm. Throughout the paper, terms i,j,k refer to indices.

### 2.2. Problem Formalization

Consider the block-wise transmission scenario of CC through the additive white Gaussian noise (AWGN) channel, see Figure 1. Prior to transmission, the message word $m\in {\left\{0,1\right\}}^{N_{m}}$ is encoded twice: By an error detection code and by an error correction code. The CRC encodes m with systematic generator matrix $G_{CRC}$. Its parity check matrix is $H_{CRC}$. We denote the detection codeword by $u\in {\left\{0,1\right\}}^{N_{u}}$ and the codebook with U. Then, the CC encodes u with generator matrix $G_{CC}$. As a result, the codeword c is a bits sequence $c=(c^{\left(1\right)},\ldots ,c^{\left(N_{u}\right)})$ with $c^{\left(i\right)}\in {\left\{0,1\right\}}^{1/R_{CC}}$ where $R_{CC}$ denotes the rate of the CC. For brevity, we denote $V=1/R_{CC}$; the length of the CC calculated as $N_{c}=N_{u}\cdot V$.

After encoding, the codeword c is BPSK-modulated (0→1, 1→−1) and x is transmitted through the channel with noise $n∼N(0,\sigma_{n}^{2}I)$. At the receiver, one decodes the LLR word *ℓ* rather than y. The LLR values are approximated based on the bits i.i.d. assumption and Gaussian prior $\ell =\frac{2}{\sigma_{n}^{2}}\cdot y$. The decoder is represented by a function $F\left(\cdot \right):R^{N_{c}}\to {\left\{0,1\right\}}^{N_{u}}$ that outputs the estimated detection codeword $\hat{u}$. Our end goal is to find u that maximizes the a posteriori optimization problem:(1)$$\hat{u}=\underset{u\in U}{arg max}P(u|\ell ).$$

Note that we solve for u rather than m as bit flips in either the systematic information bits or in the CRC bits are considered as errors.

### 2.3. Viterbi Decoding of CC

Naive solution of Equation (Equation 1) is exponential in $N_{m}$. However, it can be simplified following Bayes:(2)$$\underset{u\in U}{arg max}P(u|\ell )=\underset{u\in U}{arg max}P(u)P(\ell |u)$$
where P(ℓ) is omitted, as this term is independent of u.

The time complexity of the solution to Equation (Equation 2) is yet exponential, but may be further reduced to linear dependency in the memory’s length by following the well known Viterbi algorithm (VA) [6]. We formulate notation for this algorithm in the following paragraphs.

Denote the memory of the CC by ν and the state space by $S=\{0,\ldots ,2^{\nu }-1\}$. Convolutional codes can be represented by multiple temporal transitions, each one is a function of two arguments: the input bit and the current state. The trellis diagram is one convenient way to view these temporal relations, each trellis section is called a stage. We refer to the work in [21] for a comprehensive tutorial regarding CC.

Let the sequence of states be represented by $s\in S^{N_{u}+1}$. Following the properties of the CC, a 1-to-1 correspondence between the codeword u and the state sequence s exists:$$\underset{u\in U}{arg max}P(u)P(\ell |u)=\underset{s\in S^{N_{u}+1}}{arg max}P(s)P(\ell |s).$$

Plugging the Markov property into the previous equation leads to:$$\begin{array}{cc} \; & \underset{s\in S^{N_{u}+1}}{arg max}P(s)P(\ell |s)=\\ & \underset{s\in S^{N_{u}+1}}{arg max}\prod_{i=1}^{N_{u}}P\left(s_{i+1}|s_{i}\right)P\left(\ell_{iV-V+1;iV}|s_{i+1},s_{i}\right)=\\ & \underset{s\in S^{N_{u}+1}}{arg max}\sum_{i=1}^{N_{u}}log(P\left(s_{i+1}|s_{i}\right)+log\left(P\left(\ell_{iV-V+1;iV}|s_{i+1},s_{i}\right)\right) \end{array}$$
where the last transition is due to the monotonic nature of the log function.

Next, denote the path metric $\lambda_{i}=-log\left(P\left(s_{i+1}|s_{i}\right)\right)$ and the branch metric, representing the transition over a trellis edge, as $\beta_{i}=-log\left(P\left(\ell_{iV-V+1;iV}|s_{i+1},s_{i}\right)\right)$. Then, substituting these values into the last equation:(3)$$\underset{s\in S^{N_{u}+1}}{arg max}P(s)P(\ell |s)=\underset{s\in S^{N_{u}+1}}{arg min}\sum_{i=1}^{N_{u}}\lambda_{i}+\beta_{i}.$$

Taking a dynamic programming approach, the Viterbi algorithm solves Equation (Equation 3) efficiently:(4)$$\lambda_{i}\left(s\right)=\min_{s^{\prime }\in S}\lambda_{i-1}\left(s^{\prime }\right)+\beta_{i},\;s\in S$$
starting from i=2 up to $i=N_{u}+1$, in an incremental fashion, with the initialization:(5)$$\lambda_{1}\left(s\right)=\left\{\begin{array}{cc} -\lambda_{max} & \mathrm{if}\;s=s_{1}\\ 0 & \mathrm{otherwise} \end{array}\right.$$
and $s_{1}$ = 0. The constant $\lambda_{max}$ is called the LLR clipping parameter.

To output the decoded codeword $\hat{u}$, one has to perform the trace-back operation $\Pi :R^{N_{c}}\times S\to U$. This operation takes the LLR word along with a termination state and outputs the most likely decoded codeword: $\Pi \left(\ell ,s^{\prime }\right)=\hat{u}$. Specifically, it calculates the sequence of states $\hat{s}$ that follows the minimal $\lambda_{i}\left(s\right)$ values at each stage, starting from $s_{N_{u}+1}=s^{\prime }$ backwards. Then, the sequence $\hat{s}$ is mapped to the corresponding estimated codeword $\hat{u}$. Under the classical zero-tail termination, $\hat{u}=\Pi \left(\ell ,0\right)$ is returned.

### 2.4. Circular Viterbi Decoding of TBCC

TBCC work under the assumption of equal start and end states. Their actual values are determined by the last ν bits. As such, the MLD with a list of size 1 is the decoded u whose matching $\lambda_{N_{u}+1}\left(s^{\prime }\right)$ value is minimal, combining the decisions from multiple VA runs, one from each state $s^{\prime }$.

As mentioned in Section 1, the complexity of this MLD grows exponentially in the memory’s length. The CVA is a suboptimal decoder that exploits the circular nature of the TBCC trellis, executing VA for a specified number of repetitions, where each new VA is initialized with the end metrics of the previous repetitions. The CVA starts and ends its run at the zero state, being error prone near the zero tails.

Explicitly, the forward pass of the CVA follows Equation (Equation 4) for $i\in \{2,\ldots ,I\cdot N_{u}\}$ with *I* denoting an odd number of replications. The same initialization as in Equation (Equation 5) is used. The bits of the middle replication are the least error-prone, being farthest from the zero tails, thus returned: $${\hat{u}}_{i}={\left(\Pi \left(\ell ,0\right)\right)}_{i+\lfloor \frac{I}{2}\rfloor \cdot N_{u}}$$
for $i\in \{1,\ldots ,N_{u}\}$.

## 3. A Data-Driven Approach to TBCC Decoding

This section describes our novel approach to decoding: Parameterization of the CVA decoder, and its integration into an ensemble composed from specialized experts and a low-complexity gating.

### 3.1. Weighted Circular Viterbi Algorithm

Nachmani et al. [17] presented a weighted version of the classical Belief Propagation (BP) decoder [22]. This learnable decoder is the deep unfolding of the BP [23]. This weighted decoder outperforms the classical unweighted one by training over channel words, adjusting the weights to compensate for short cycles that are known to prevent convergence.

We follow the favorable model-based approach as well, parameterizing the branch metrics that correspond to edges of the trellis. We add another degree of freedom for each edge, assigning weights to the path metrics as well. Considering the complexity overhead, we only parameterize the middle replication. This formulation unfolds the middle replication of the CVA as a Neural Network (NN):(6)$$\lambda_{i}\left(s\right)=\min_{s^{\prime }\in S}w_{i,s^{\prime },s}\lambda_{i-1}\left(s^{\prime }\right)+w_{i,\beta }\beta_{i}$$
for $\lfloor \frac{I}{2}\rfloor \cdot N_{u}\leq i\leq \lceil \frac{I}{2}\rceil \cdot N_{u}$.

Our goal is to calculate parameters $\left\{w_{i,s,s^{\prime }},w_{i,\beta }\right\}$ that achieve termination states equal to the ground-truth start and end states. The exact equality criterion is non-differentiable; thus, we minimize the multi-class cross entropy loss, acting as a surrogate loss [24]:$$L\left(s,\lambda \right)=-\log\sigma (\lambda^{l}\left(s_{N_{u}+1}\right))$$
where $\lambda^{l}\left(\cdot \right)=\lambda_{\lceil \frac{I}{2}\rceil \cdot N_{u}}\left(\cdot \right)$ stands for the last learnable layer, and σ being the softmax function:$$\sigma \left(\lambda^{l}\left(s\right)\right)=\frac{e^{\lambda^{l}\left(s\right)}}{\sum_{s^{\prime }\in S}e^{\lambda^{l}\left(s^{\prime }\right)}}$$

This specific choice encourages the equality of the end states in the mid-replication to their ground-truth values. Note that the gradients back-propagate through the non-differentiable min criterion in Equation (Equation 6) as in the maximum pooling operation: They only affect the state that achieved the minimum metric.

One fallacy of this approach is the similar importance for all edges, contrary to the BP, where not all edges are created equal (e.g., ones that participate in many short cycles). All edges are of the same importance as derived from the problem’s symmetry: Due to the unknown initial state, each state is equally likely.

This indicates that training this architecture may leave the weights as they are, or at worst even lead to divergence. To fully exploit the performance gained by the adjustment of the weights, one must first break the symmetry. We alter the uniform prior over the termination states by assigning only a subset of the termination states to a single decoder. We further elaborate on this proposition below.

### 3.2. Ensembles in Decoding

Ensembles [25,26] shine in data-driven applications: They exploit independence between the base models to enhance accuracy. The expressive power of the ensemble surpasses that of a single model. Thus, whereas a single model may fail to capture high-dimensional and nonlinear relations in the dataset, a combination of such models may succeed.

Nonetheless, ensembles also encompass computational complexity which is linear in the number of base learners, being unrealistic for practical considerations. To reduce complexity, our previous work [20] suggests to employ a low complexity gating decoder. This decoder allows one to uniquely map each input word to a single most fitting decoder, keeping the overall computation complexity low. We further elaborate on the gated ensemble, referred to as gated WCVAE, in Section 3.3 and gating in Section 3.4.

### 3.3. Specialized-Experts Ensemble

The WCVAE is an ensemble comprised of WCVA experts, each one specialized on words from a specific subset of termination states. We begin by discussing the forming of the experts in training, see Figure 2 for the relevant flowchart.

### 3.4. Gating

Let the number of trainable WCVA decoders in the WCVAE be α, with each decoder possessing *I* repetitions. To form the experts, we first simulate many message words randomly, each message word is encoded and transmitted through the channel. The initial state of a transmitted word u, known in training, is denoted by $s_{1}$ as before. Then, we add the tuple (ℓ,u) to the dataset representing the subset of states that include state $s_{1}$:(7)$$D^{\left(i\right)}=\left\{\left(\ell ,u\right):\frac{2^{\nu }}{\alpha }\cdot \left(i-1\right)\leq s_{1}\leq \frac{2^{\nu }}{\alpha }\cdot i-1\right\}$$
with i∈{1,…,α} and ν is yet the memory of the CC. Each dataset accumulates a high number of words by the above procedure.

All the WCVA decoders are trained as in the guidelines of Section 3.1, with one exception: The *i*th decoder is trained with the corresponding $D^{\left(i\right)}$. Subsequently, α specialized experts are formed, each one specializes on decoding words affiliated to a specific subset of termination states. Each codeword has equal probability to appear, thus the distribution over the termination states is uniform. We further elaborate on the intuition to this particular division of close-by states in Section 4.4.

One common practice is to separate TBCC decoding into an initial state estimation followed by decoding. For example, Fedorenko et al. [27] run a soft-input soft-output (SISO) decoder prior to LVA decoding. This prerun determines the most reliable starting state.

Similarly, our work presents a gating decoder which acts as a coarse state estimation. The gating is composed of two parts: a single forward pass of the CVA and a multiple trace-backs phase. We only employ the gating in the evaluation phase; Check Figure 3 for the complete flow.

First, a forward pass of a CVA is executed on the input word *ℓ*, as in Equation (Equation 4). As all states are equiprobable, the initialization is chosen as $\lambda_{1}\left(s\right)=0,\forall s\in S$ instead of Equation (Equation 5). After calculating $\lambda_{i}\left(s\right)$ for every state and stage, the trace-back Π(·) runs α times, each time starting from a different state. The starting states are spread uniformly over S, with the decoded words given as
(8)$${\tilde{u}}^{\left(i\right)}=\Pi \left(\ell ,\frac{2^{\nu }}{\alpha }\cdot \left(i-\frac{1}{2}\right)\right),1\leq i\leq \alpha .$$

Notice that the trace-back is a cheap operation, compared to the forward pass [7]. Next, the value of the CRC syndrome is calculated for each trace-back: (9)$$g_{i}=\|{\tilde{u}}^{\left(i\right)}H_{CRC}^{T}\|$$
with $g_{i}=0$ indicating that no error has occurred (or is detectable). In case that a single $g_{i}$ is zero, the corresponding decoded word ${\tilde{u}}^{\left(i\right)}$ is output. If more than one $g_{i}$ is zero, the decoded word is chosen randomly among all candidates. Only if no *i* exists such that $g_{i}=0$, the word *ℓ* continues for additional decoding at the ensemble.

Note that each computed value $g_{i}\neq 0$ is correlative with the ascription of the word *ℓ* to the termination state $\frac{2^{\nu }}{\alpha }\cdot \left(i-\frac{1}{2}\right)$. Though the ground-truth termination state may not necessarily have the minimum $g_{i}$, this minimal value still hints that the ground-truth state is close by. As a result, decoding with the decoder corresponding to the minimal $g_{i}$ is satisfactory. If multiple $g_{i_{1}},\ldots ,g_{i_{k}}$ share the same minimal value then decoders $i_{1},\ldots ,i_{k}$ decode the word, choosing the output ${\hat{u}}^{\left(i\right)}$ of minimal CRC value among the candidates.

## 4. Results

### 4.1. Performance and Complexity Comparisons

The WCVAE was simulated with CRC codes and TBCC that are in accordance with the LTE standard. Note that while LTE employs QPSK modulation, we used BPSK for simplicity. A code of specific length is denoted with ($N_{c},\;N_{u},\;N_{m}$), referring to the code’s length, detection codeword’s length, and message’s length, respectively. A summary of relevant code parameters appears in Table 1.

We compared both the gated WCVAE and the WCVAE to the next common baselines:**3-repetition CVA**—a fixed-repetitions CVA [7].**List circular Viterbi algorithm (LCVA)**—an LVA that runs CVA instead of a VA; All other details are as explained in Section 1.**List genie VA (LGVA)**—an LVA decoder with list of size α, that runs from a known ground-truth state; The optimal decoded codeword is chosen by the CRC criterion. The FER of the gated and non-gated WCVAE are lower bounded by this genie-empowered decoder.

All Monte Carlo experiments ran on a validation dataset composed of signal-to-noise ratio (SNR) values in the range of −2 dB to 2 dB with a step of 1 dB. Simulations at each point continued until at least 500 errors were accumulated. The number of decoders was set to α=8. As words are drawn from the channels arbitrarily, the notion of “epoch” which refers to the number of full transitions over the training dataset is ill defined: We instead provide the number of training mini-batches. All decoders, i.e., the gating and the experts, were executed with I=3 repetitions. The overall hyperparameters for the ensemble training are depicted in Table 2.

Figure 4 presents the results for the two different lengths: Both in error rate and computational complexity (measured in VA runs). The method achieves FER gains of up to 0.75 dB and 0.625 dB gain over the CVA in the waterfall region, for the lengths 13 and 15, respectively. Our method also surpasses the LCVA by a small margin. Considering the complexity of the scheme, the number of VA runs decreases as a function of the SNR and converges with the 3-repetition CVA in high SNR values. As the trace-back has negligible complexity compared to the forward pass of the VA [7], one may claim that the computational complexities at evaluation are similar.

### 4.2. Generalization to Longer Lengths

As mentioned in Section 1, one benefit of the model-based approach is the capability to easily generalize to longer codes. This benefit should apply to our proposed method; To test this notion, we trained and evaluated the WCVAE on the same code but over two longer lengths. All other codes parameters and training hyperparameters are exactly as in Table 1 and Table 2.

Figure 5 depicts the performance over two longer codes. Notice that the gain is around the 0.6 dB in FER, similarly to the one on the two shorter codes. This empirically shows the generalization of the novel method to longer lengths. The training process remains as simple as before, even as the length increases; There is no need to enforce a curriculum based ramp-up method for convergence as in [14].

### 4.3. Training Analysis

We provide further insights to the benefits of training by studying the performance of the trained specialized decoders versus their non-trained counterparts. We fixed the code to TBCC (87,29,13) and the SNR to 0 dB. Figure 6 depicts the FER as function of the termination states, each subplot shows two decoders: The classical CVA and the trained WCVA. The *i*th classical CVA had 3 repetitions, as before, and ran trace-back from state $\frac{2^{\nu }}{\alpha }\cdot \left(i-1\right)$. The trained WCVA is the *i*th decoder of the WCVAE, responsible for decoding states $\left\{\frac{2^{\nu }}{\alpha }\cdot \left(i-1\right),\ldots ,\frac{2^{\nu }}{\alpha }\cdot i-1\right\}$. It ran trace-back from the same state. At each point, codewords of the given state, and **only this state**, were simulated until 250 accumulated errors.

One may observe that the CVA has peak performance at the trace-back state, yet at all other states it performs poorly. On the other hand, the WCVA decoders manage a trade-off: They sacrifice performance over the trace-back state, compensating for this loss by achieving lower error at other states. To conclude this part, note the specialized decoders indeed specialize at decoding words with a termination state included in their respective subset of termination states.

### 4.4. Ensemble Size Evaluation

In Figure 7, we inspect the performance of our method over different ensemble sizes, α∈{4,8,16,32}. An ensemble with α decoders is denoted by α-WCVAE. We fixed the code to TBCC (87,29,13) and simulated each ensemble with 500 accumulated errors (per point). All other parameters and hyperparameters are the same as in Table 1 and Table 2.

The figure implies that increasing the number of decoders by a factor of two results in around 0.1 dB FER gain. This simulation empirically validates an intuitive assumption: Our ensemble is more diverse as its size increases, i.e., it successfully captures a larger portion of the input space.

## 5. Discussion

This work follows the model-based approach and applies it for TBCC decoding, starting with the parameterization of the common CVA decoder. Its parameterization relies on domain knowledge to effectively exploit the decoder: A classical low-complexity CVA acts as a gating decoder, filtering easy to decode channel words and directing harder ones to fitting experts; Each expert is specialized in decoding words that belong to a specific subset of termination states. This solution improves the overall performance, compared to a single decoder, as well as reduces the complexity in a data-driven fashion.

Future directions are to extend the ensemble approach to more use cases, such as different codes and various learnable decoders. For example, an ensemble tailored for polar codes, with a CRC-based gating, is one idea we intend to explore. This scenario is indeed practical, as 5G standard incorporates polar codes accompanied by CRC codes. Another direction is the theoretical study and analysis of the input space, e.g., the regions of the pseudo-codewords [5] and tailbits errors [7]. This could direct the training of the learnable decoder to surpass current results.

## Figures and Tables

**Figure 1 entropy-23-00093-f001:**
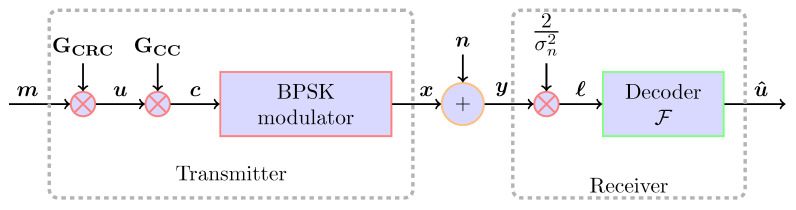
System diagram.

**Figure 2 entropy-23-00093-f002:**
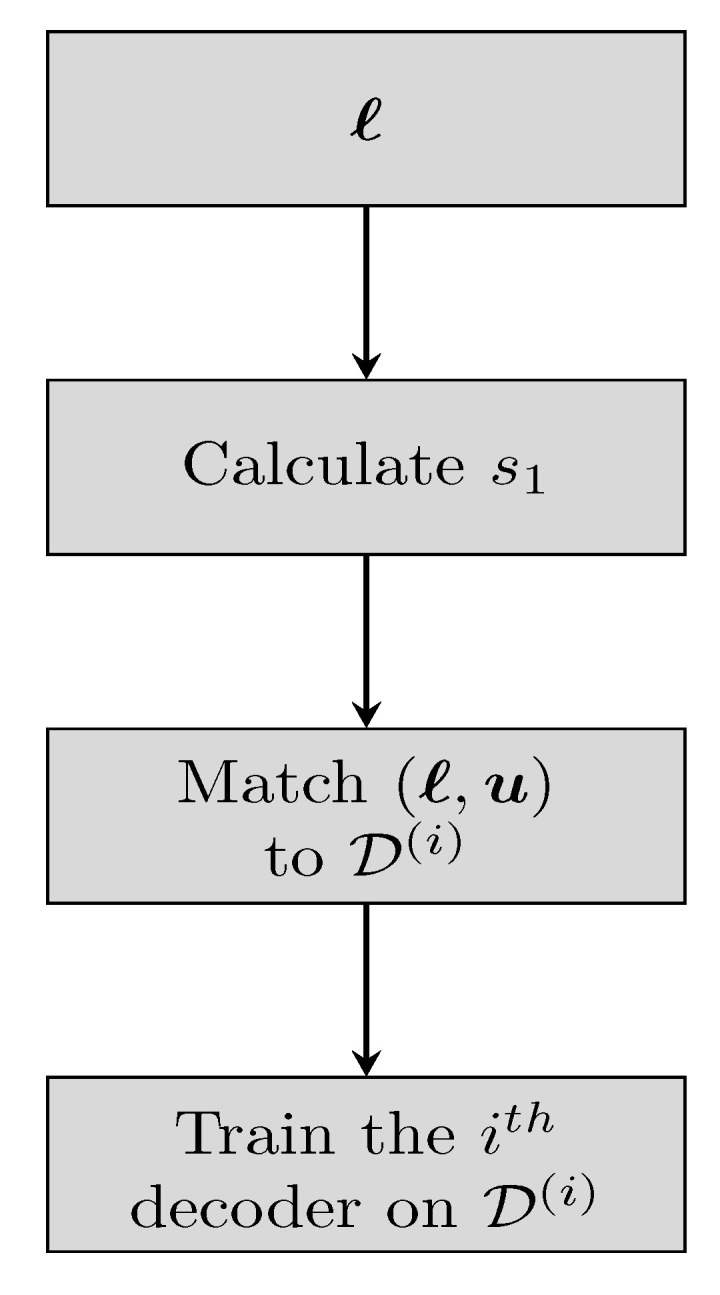
Training flowchart.

**Figure 3 entropy-23-00093-f003:**
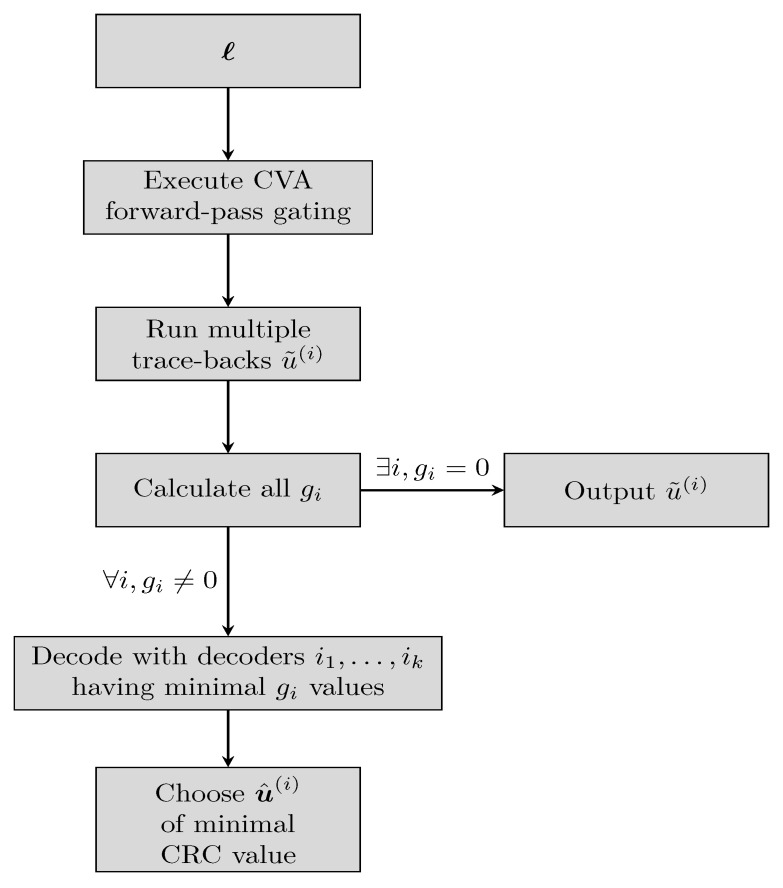
Evaluation flowchart.

**Figure 4 entropy-23-00093-f004:**
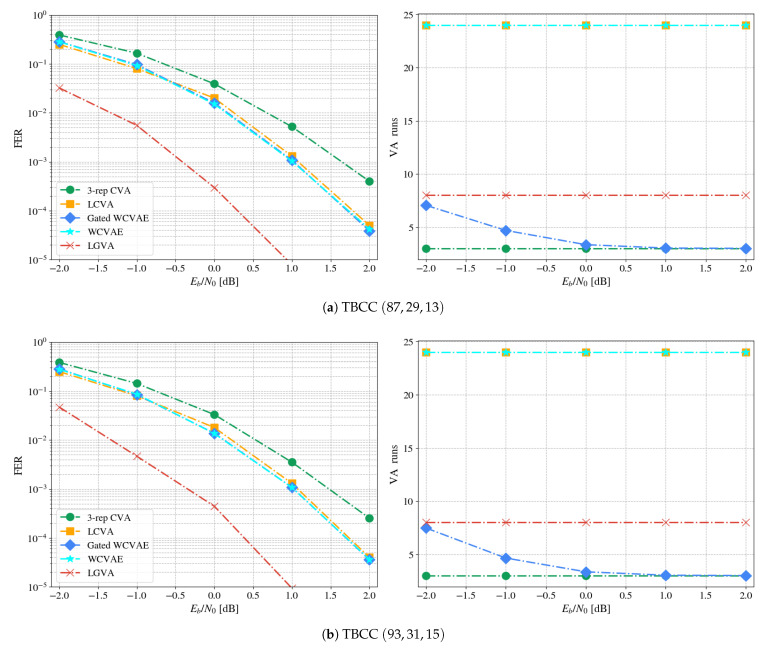
FERand complexity plots for decoding LTE-TBCC.

**Figure 5 entropy-23-00093-f005:**
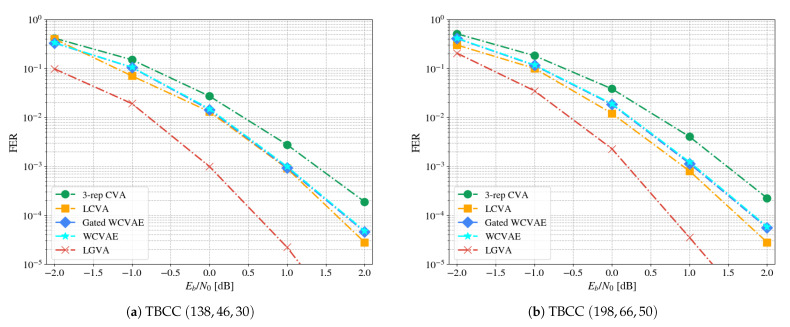
Generalization to longer lengths.

**Figure 6 entropy-23-00093-f006:**
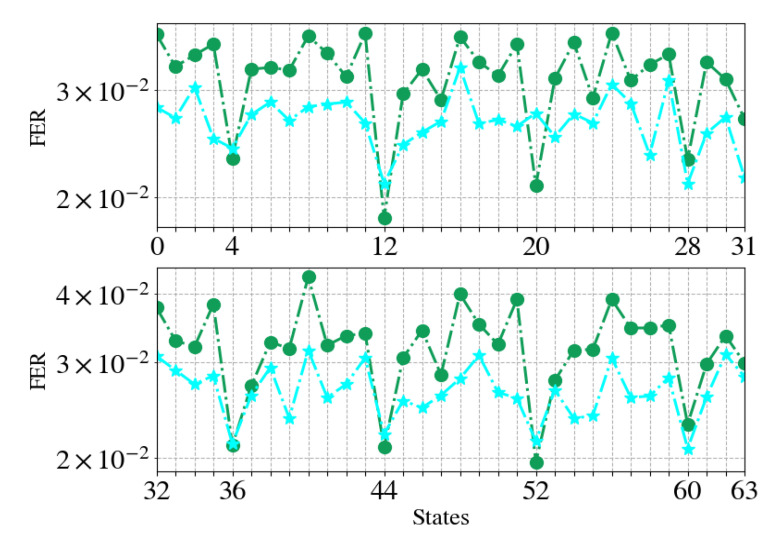
Training analysis. Legend: (•) CVA, (★) WCVA.

**Figure 7 entropy-23-00093-f007:**
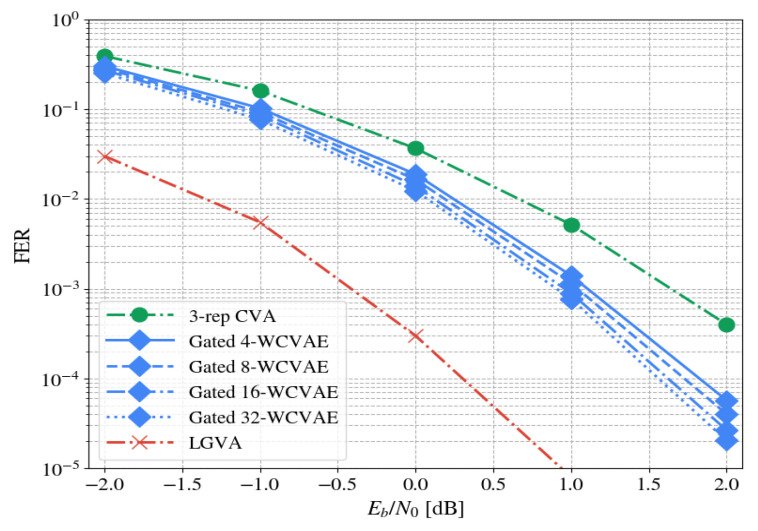
The effect of different ensemble sizes on performance.

**Table 1 entropy-23-00093-t001:** Code parameters

Symbol	Definition	Value
ν	CC memory size	6
-	CC polynomials	(133,171,165)
$R_{CC}$	CC rate	1/3
-	CRC length	16

**Table 2 entropy-23-00093-t002:** Hyper-parameters of the ensemble

Symbol	Definition	Value
α	Ensemble size	8
*I*	Repetitions per decoder	3
$\lambda_{max}$	LLR clipping	20
lr	Learning rate	$10^{-3}$
-	Optimizer	RMSPROP
-	Loss	Cross Entropy
-	Training SNR range [dB]	(−2)–0
-	Mini-batch size	450
-	Number of mini-batches	50

## Data Availability

Data simulation is found in the github repository.

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
