# Peer review of "Deep Ensemble of Weighted Viterbi Decoders for Tail-Biting Convolutional Codes"

_entropy, 2021, doi:10.3390/e23010093_

Round 1

Reviewer 1 Report

The idea presented in this paper is interesting. I have a few minor concern as following.

  1. Fig.6,  please show the definition of Y-axis
  1. page 7, Eq(9), H_CC should be H_CRC
  1. page 7, line 193 needs more explanation. In case all paths failed passing CRC, the path corresponding to the ground truth termination state may not necessarily have the minimum Hamming weight for the CRC syndrome.
  1. I hope the authors could compare TBCC/Gated WCVAE with punctured CC/VA. For example, the FER of CC with Nm=50, Nu=50+16 crc bits + 6 tail bits, N=3*Nu-18 punctured bits =198 code bits can be added to Fig. 5(b).

Reviewer 2 Report

The authors apply ensemble approach in decoding TBCC. Each decoder is trained with a subset of TBCC (l,u) that starts/ends with a subset of states. The authors show that the FER of the list circular Viterbi Algorithm can be met by the proposed ensemble decoder, which requires smaller number of VA runs (as shown in Figure 4). Overall, the idea and the message are clear. Some suggestions and comments are as follows. 

  • Ablations study with varying values of alpha (number of decoders) will be interesting.
  • FER values (Y label) are missing in Figure 6. 
  • I might have misunderstood. By Gini, do you mean genie? 
  • Notations are often confusing. For example, I had to go back to find the meaning of k. In figure 3, the authors write "Decode with decoders i1,...ik having minimal gi values. There is only one minimal gi. also had to look back to clarify \nu when I was reading (7). 

Round 2

Reviewer 1 Report

I’m satisfied with the revised version.